# Disseminated *Toxoplasma gondii* Infection in an Adult Osprey (*Pandion haliaetus*)

**DOI:** 10.3390/vetsci9010005

**Published:** 2021-12-24

**Authors:** Xiaobo Wang, Charles T. Talbot, Ji-Hang Yin, Anwar A. Kalalah, Chengming Wang, Joseph C. Newton

**Affiliations:** 1College of Veterinary Medicine, Yangzhou University; Jiangsu Co-Innovation Center for the Prevention and Control of Important Animal Infectious Disease and Zoonoses, Yangzhou University, Yangzhou 225012, China; wangxb@yzu.edu.cn; 2Department of Pathobiology, College of Veterinary Medicine, Auburn University, Auburn, AL 36849, USA; charlie.talbot5@gmail.com (C.T.T.); jzy0089@auburn.edu (J.-H.Y.); anwarabdulaziz.kalalah@utsa.edu (A.A.K.); wangche@auburn.edu (C.W.)

**Keywords:** *Toxoplasma gondii*, osprey, *Pandion haliaetus*, histopathology, IHC

## Abstract

An adult female osprey (*Pandion haliaetus*) was found weak and unable to fly in Auburn, Alabama in August 2019. The bird was captured and submitted to the Southeastern Raptor Center of the Auburn University College of Veterinary Medicine for evaluation. On presentation, the bird was thin with a body condition score of approximately 1.5 out of 5. The bird died during the examination and was submitted for necropsy. At the necropsy, there was a severe loss of muscle mass over the body, and the keel was prominent. The liver and spleen were moderately enlarged with pale tan to red foci randomly scattered throughout the parenchyma. A histopathologic observation revealed multifocal to coalescing areas of necrosis and hemorrhage with intralesional protozoans in the liver, spleen, lungs, kidney, sciatic nerve, esophagus, cerebrum, heart, and proventriculus. Immunohistochemistry using anti-*Toxoplasma gondii*-specific antibodies showed a strong positive labeling of the parasite. Semi-nested PCR, specific for the B1 gene of *T. gondii*, successfully identified *T. gondii*. This is the first confirmed case of *T. gondii* infection in an osprey.

## 1. Introduction

*Toxoplasma gondii* (genus Apicomplexa: family Sarcocystidae) is a protozoan parasite infective to a wide range of wild and domestic, warm-blooded animals including birds and human beings [1]. The only known definitive hosts of the parasite are wild and domestic felids [2]. In felids, following the ingestion of oocysts or infected tissue from an intermediate host, the parasite undergoes sexual reproduction in the intestinal epithelium, producing millions of oocysts that are shed in the feces [3]. All other affected animals, including numerous mammals and birds, are considered intermediate hosts, and become infected via the consumption of oocysts in food and/or water contaminated with fecal matter from felids, or by the consumption of animal products containing encysted stages (bradyzoites) of the parasite [2,3,4]. Within intermediate hosts, the parasite undergoes only asexual reproduction [5]. Following the ingestion of oocysts or bradyzoite cysts by an intermediate host, sporozoites or bradyzoites are liberated in the intestine where they penetrate the intestinal epithelium and differentiate into tachyzoites. Tachyzoites are disseminated via the bloodstream throughout the animal within leukocytes and rapidly replicate intracellularly within a large variety of cell types, causing tissue damage and necrosis [6]. Young and immunocompromised animals may succumb to generalized toxoplasmosis at this stage [7]. Older, immunocompetent animals have a robust, cell-mediated immune response to the tachyzoites, and the infection becomes latent [8]. In these intermediate hosts, the tachyzoites transform into encysted bradyzoites, which can remain viable in a variety of tissues for many years, possibly for the life of the animal [9].

## 2. Case History

An adult female osprey was found weak and unable to fly in Chewacla State Park of Auburn, Alabama, USA. The bird was captured and presented to the Southeastern Raptor Center of the Auburn University College of Veterinary Medicine for evaluation. A physical examination was initiated upon presentation, but the osprey died during the procedure. The body was submitted for necropsy and a postmortem examination was conducted a few hours after death.

At necropsy, the osprey was in a poor body condition (body condition score of 1.5/5) with mild postmortem changes. There was a severe loss of muscle mass over the body and the keel was prominent. The liver was moderately enlarged with multifocal to coalescing areas of hemorrhage scattered over the capsular surface. Multifocal to coalescing, 1 mm × 1-mm, pale-brown-to-yellow, well-demarcated, smooth foci extending from Glisson’s capsule into the parenchyma were observed throughout the liver (Figure 1a). The spleen was moderately enlarged, diffusely pale, and firm with a multifocal, 1 mm × 2 mm, dark tan and smooth, circular foci that extended into the parenchyma from the splenic capsule (Figure 1b). The respiratory system, urinary system, integumentary system, endocrine system, heart, and brain were grossly unremarkable. Portions of liver and spleen were submitted for aerobic bacterial and fungal culture and a light growth of *Edwardsiella tarda* was obtained from both organs. Fungal cultures were negative. An oral swab was negative for avian influenza virus and no West Nile virus was detected in the submitted cerebrum. Tissue samples from the cerebrum, liver, spleen, lungs, kidneys, heart, sciatic nerve, stomach, duodenum, pancreas, jejunum, ileum, colon, cecum, esophagus, and trachea were fixed in 10% neutral-buffered formalin and paraffin processed for routine histopathologic evaluation. Four micron sections were cut and stained with hematoxylin and eosin (H&E) for microscopic examination. Sections of the liver and spleen were additionally processed for anti-*T. gondii* immunohistochemistry.

Histologically, approximately 60% of the hepatic parenchyma was effaced by multifocal to coalescing areas of hepatocellular necrosis characterized by the loss of tissue architecture with replacement by pale, granular eosinophilic cellular debris admixed with viable and degenerative heterophils, macrophages, lymphocytes, a few plasma cells, and extravasated erythrocytes. Similar inflammatory cells cuffed the hepatic vasculature. Associative macrophages contained numerous intracytoplasmic, round, basophilic, 2-μm protozoa with indistinct internal structures (Figure 2a,b). These organisms, histologically consistent with *T. gondii*, were also found in the surrounding extracellular spaces. Immunohistochemical staining, using a rabbit polyclonal antibody against *T. gondii* as a primary antibody and a horse anti-rabbit horseradish peroxidase as a secondary antibody, were positive for the parasite, with membranous immunoreactivity to the intracytoplasmic tachyzoites. Replacing most of the splenic parenchyma were multifocal-to-coalescing areas of necrosis characterized by pale, granular, eosinophilic material, with cellular debris, extravasated erythrocytes, fibroblasts, degenerative heterophils, lymphocytes, and plasma cells. Macrophages and adjacent clear spaces contained bradyzoites and tachyzoites of *T. gondii* (Figure 2c). Multifocally, and throughout the congested lung parenchyma were variably sized foci of necrosis containing the protozoal organisms noted above (Figure 2d). The myocardium had multifocal areas of myocardiocyte degeneration and necrosis characterized by the fragmentation and vacuolation of the myocardial fibers accompanied by the accumulation of heterophilic and lymphocytic inflammatory cells, with scant numbers of macrophages containing protozoal organisms. The muscularis externa of the proventriculus contained multifocal aggregations of lymphocytes and plasma cells, with few heterophils, and protozoal-laden macrophages. Similar aggregations of inflammatory cells were observed in the surrounding vessels within the serosal adipose tissue. Infiltrating and expanding the epineurium of sections of the sciatic nerve were dense populations of lymphocytes and plasma cells, with multifocal areas of necrosis containing small numbers of protozoal-laden macrophages. Previously described inflammatory cells were also present, partially cuffing the esophageal mucosal salivary glands and the serosal vasculature. The renal vasculature and cortical and medullary interstitium were moderately congested. Similar tachyzoites were noted in the cerebrum, esophagus and proventriculus. Tissue sections of the brain stem and cerebellum were histologically unremarkable.

The presence of *T. gondii* DNA in tissues was assessed by qPCR protocols targeting a specific 130 base pair sequence of the B1 gene and a 529 base pair sequence of the repetitive element (529 bp-RE) as previously described [10,11]. DNA was extracted from formalin-fixed, paraffin-embedded (FFPE) tissue as previously described [12]. Oligonucleotide primers were used to amplify regions of the B1 gene of *T. gondii*: TOXO1 (5′-GGAACTGC ATCCGTTCATGAG-3′) and TOXO2 (5′-TCTTTAAAGCGTTCGTGGTC-3′). The repetitive region of *T. gondii* was amplified with the oligonucleotide primers: TOX4 (5′-CGCTGCAGGGAGGAAGACGAAAGTTG-3′) and TOX5 (5′-CGCTGCAGACAGAGTGCATCTGGATT-3′). The BLASTn demonstrated the identified *T. gondii* from this work had an 100% identity with the *T. gondii* isolate EIReal109a (GenBank Accession #: KX270373.1; an isolate from a naturally infected sheep in Mexico), and the *T. gondii* VEG strain (GenBank Accession #: LN714499.1).

Based on the gross and histopathological findings and molecular analyses, the bird was diagnosed with disseminated toxoplasmosis. Although *T. gondii* was isolated from other raptors in the US [2,13], to our knowledge, this is the first reported clinical case of toxoplasmosis from an osprey.

## 3. Discussion

Ospreys are diurnal, migratory birds of prey with a worldwide distribution, wintering or breeding on every continent except Antarctica [14]. The bird is the sole species within the family Pandionidae. Four subspecies, separated by geographic region, are recognized, including *Pandion haliaetus carolinensis*, *P. haliaetus haliaetus*, *P. haliaetus rudgayi* and *P. haliaetus leucocephalus* [14]. *P. haliaetus carolinensis*, the subspecies found in the southeastern United States, breeds across North America and the Caribbean, and winters in Central and South America. In North America, their range extends from much of Alaska and Canada downward throughout all regions of the continental United States and Mexico. There are resident populations in the southeastern United States and Southern California that do not migrate but spend their entire year in those locations [14,15].

Serologic surveys indicate that an exposure to *T. gondii* in wild birds is common, and viable *T. gondii* bradyzoites were isolated from the tissues of a variety of raptor species [13,16,17]. However, clinical toxoplasmosis in birds, including wild raptors, is rare [1,2]. Myocarditis with intralesional *T. gondii* organisms was reported in a bald eagle in New Hampshire [16] and necrotizing hepatitis with intralesional *T. gondii* organisms was observed in an adult barred owl in Canada [18]. In a study examining the prevalence of encysted *T. gondii* in raptors in the southeastern US, the researchers isolated viable *T. gondii* from 27 (26.7%) of the 101 raptors surveyed [13]. Four ospreys comprised the study group and all four were negative for the parasite. In a second study of southeastern raptors, the prevalence of antibodies to *T. gondii* in 281 birds, in a raptor rehabilitation facility between 2012 and 2014, was examined. This survey found a variety of hawks, owls, and eagles to be seropositive for *T. gondii*, but American kestrels, Mississippi kites and an osprey were serologically negative [17]. A study in Europe describing the prevalence and risk factors associated with *T. gondii* infection in wild birds found 29% of surveyed ospreys to be seropositive [19]. The data indicated the main risk factors associated with *T. gondii* seropositivity in all wild birds tested were bird age and feeding behavior, with the highest exposure observed in older animals and in species with a carnivorous diet. Fish-eating birds were much less likely to have serum antibodies against *T. gondii* than the carnivorous species. In experimental infections of raptors with the parasite, clinical disease was not observed [2,20,21].

The source of *T. gondii* infection in the present case is unknown. The bird was likely infected with the protozoan through the ingestion of oocysts from the soil or water or by the ingestion of intermediate hosts harboring encysted bradyzoites in their tissues. Additionally, the infection of the bird could have occurred secondary to the ingestion of oocysts present in the gastrointestinal tract of fish. Oocyst ingestion is a common source of infection in a variety of species, including humans [3]. The extremely resistant oocysts of *T. gondii* can remain viable for over a year in soil [22], freshwater [23], and saltwater [24], and the infectious dose may be as low as a single oocyst [2]. Ospreys are almost exclusively piscivorous raptors with fish comprising approximately 99 percent of their diet. Occasionally these birds may prey on rodents, rabbits, hares, other birds, and small reptiles [14], which are common intermediate hosts for *T. gondii* [25]. This suggests that these small animals may have played a role in the infection of this osprey. The importance of fish-derived *T. gondii* as a cause of toxoplasmosis in this bird is unknown but fish were implicated as a source of *T. gondii* infection for human beings and marine mammals [26]. Fish are not generally considered competent biological hosts for *T. gondii* but can be contaminated with *T. gondii* oocysts from water run-off following rains and flooding, and act as mechanical carriers for the parasite [27]. Previous studies of *T. gondii* in fish and invertebrates focused on the possibility of these aquatic animals being a source of *T. gondii* infection for marine mammals and examined the concentration and survival of protozoal stages in fish and invertebrates [26,28]. *T. gondii* was shown to persist in many invertebrates [29,30,31,32], and some fish [28], but the proliferation of the parasite was not described in those animals which had a variable body temperature (poikilotherm). Experimentally infected goldfish (*Carassius auratus*), maintained at 37 °C with *T. gondii*, failed to contract *T. gondii* infections [33]. However, *T. gondii* tachyzoites were demonstrated in zebrafish (*Danio rerio*) infected experimentally with bradyzoites derived from mouse brain, suggesting that the development of *T. gondii* is possible in fish [34]. The presence of *T. gondii* in fish in the southeastern United States is unknown and the importance of fish in the transmission of the parasite to this osprey remains in question.

## 4. Conclusions

In the present study, we report the first case of disseminated *T. gondii* infection with severe, widespread tissue necrosis in an osprey.

## Figures and Tables

**Figure 1 vetsci-09-00005-f001:**
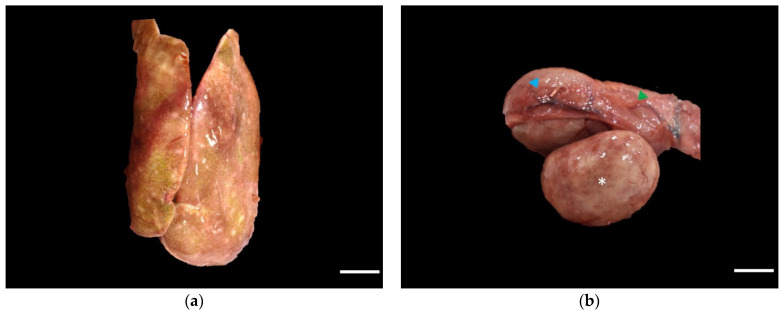
Gross images of affected liver (**a**) and spleen ((**b**), *). The liver contains multifocal to coalescing, well-demarcated, smooth foci; scale bar = 1 cm (**a**). The spleen is enlarged (*), has a diffusely pale tan and is firm with multifocal, smooth, and dark red circular foci; scale bar = 1 cm. Proventriculus (green arrowhead) and ventriculus (blue arrowhead) (**b**).

**Figure 2 vetsci-09-00005-f002:**
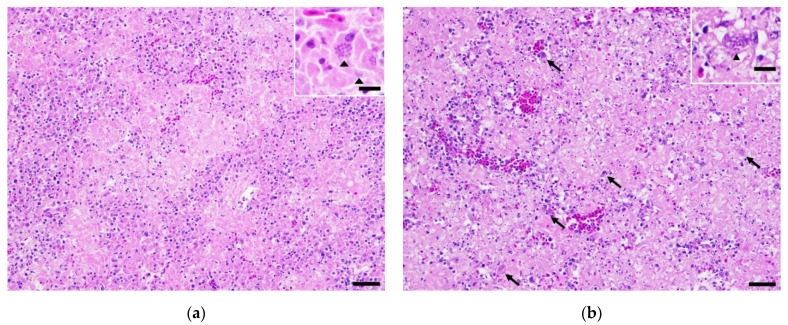
Photomicrograph of liver showing multifocal to coalescing hepatocellular necrosis. H&E. Original magnification 200X. Scale bar = 50 µm. Insert shows parasitic cyst containing bradyzoites of *T. gondii* (arrowhead). H&E. Original magnification 600X. Scale bar = 20 µm (**a**). Photomicrograph of spleen showing severe multifocal to coalescing necrosis with intralesional Toxoplasma tachyzoites (arrow). H&E. Original magnification 200X. Scale bar = 50 µm. Insert shows parasitic cyst containing bradyzoites of *T. gondii* (arrowhead). H&E. Original magnification 600X. Scale bar = 20 µm (**b**). Toxoplasma immunohistochemical staining of splenic tissue with multifocal, positively staining, intracellular protozoa (arrowhead). H&E. Original magnification 600X. Scale bar = 20 µm (**c**). Toxoplasma immunohistochemical staining of lung with positively staining intracellular protozoa (arrowhead). H&E. Original magnification 600X. Scale bar = 20 µm (**d**).

## Data Availability

The data presented were obtained from all subjects involved in this study.

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
