# Peer review of "Disseminated Toxoplasma gondii Infection in an Adult Osprey (Pandion haliaetus)"

_vetsci, 2021, doi:10.3390/vetsci9010005_

Round 1

Reviewer 1 Report

This is very interesting paper.  One of my first questions is why was this isolate not genotyped so that can be placed in publication for other researchers to know?

See attached document with comments and edits to be used in your revisions 

Author Response

Point 1: This is very interesting paper.  One of my first questions is why was this isolate not genotyped so that can be placed in publication for other researchers to know?

Response 1: Thank you. For the question of genotyping, in order to support the result of histopathological and immunohistochemistry observations, the samples of the raptor were submitted to the molecular diagnostic lab for PCR, and we found B1 gene and 529 base pair sequence of the repetitive element are conservative gene through searching some references. We design primer of B1 gene and 529 base pair sequence of the repetitive element according to the references. Although we had the result of PCR, we could’t genotyped the toxoplasma based on the B1 gene and 529 base pair sequence of the repetitive element; however, the PCR of B1 gene and 529 base pair sequence of the repetitive element could confirm our diagnosis.

Point 2: This is a super cool case report! Were you able to genotype it? if not we would love to help with that!

Response 2: Response: Till now, we couldn’t genotype the toxoplasma gondii using the B1 gene and 529 base pair sequence of the repetitive element. Indeed, we appreciate to genotype it if you are able to help with that. Thank you.

Point 3:

Page 1 line 17

Response 3: The “swollen” has been changed to enlarged, and the “and” has been changed to with.

Point 4:

Page 1 line 31, 32

Very wordy and hard to follow -make sure to specify that ingested tissue has to be infected with cysts present.

Response 4: The sentence has been revised that "After ingestion of oocysts or infected tissue from an intermediate host, the parasite undergoes sexual reproduction in the intestinal epithelium producing millions of oocysts that are shed in the feces"

Point 5:

Page 1 line 37

Response 5: The reproduction has been added.

Point 6:

Page 1 line 45

Response 6: The “is controlled” has been changed to “becomes latent”

Point 7:

Page 2 line 57

Response 7: The “swollen” has been changed to "enlarged".

Point 8:

Page 2 line 58

Response 8: The “diffuse and” have been deleted.

Point 9:

Page 2 line 60

Response 9: the downward has been deleted and the “over the hepatic parenchyma” has been changed to “throughout the liver”.

Point 10:

Page 2 line 61

Response 10: The “swollen” has been changed to "enlarged".

Point 11:

Page 2 line 66

What is the significance of this? It is associated with septicemia in aquatic animals, secondary septicemia due to toxo infection, or is it a commensal in osprey?

Response 11: Yes, this is a good question. Since Edwardsiella tarda, a member of the family Enterobacteriaceae, is a motile, facultatively anaerobic, Gram-negative rod that has been isolated from fresh and brackish water environments and a variety of animals (reptiles, amphibians, and fish, including catfish and eels). In humans, E. tarda is a rare pathogen and mainly causes gastroenteritis. Till now, there is no reference that approve the relationship or it is a commensal in osprey. However, the bacterial and the fungal culture are the standard necropsy process at Auburn University. Indeed, this is an interesting subject, we need do more further research.

Point 12:

Page 2 line 88 Is this an indicator of chronicity?

Response 12: Collagen is not an indicator of chronicity, and the description has been discarded.

Point 13:

Page 2 line 95

Response 13: the “mucularis” has been changed to Muscularis.

Point 14:

Page 3 line 100,101 hard to follow,

Response 14: the sentence has been revised. “Previously described inflammatory cells were also present partially cuffing the esophageal mucosal salivary glands and the serosal vasculature.”

Point 15:

Page 3 line 109 How long were these in fixed and embedded? The DNA is degraded over time.

Response 15: The samples were fixed in the formalin immediately after the necropsy, and the tissues were trimmed, processed and embedded the next day. The DNA extraction from formalin-fixed, paraffin-embedded (FFPE) tissues were five days after the tissue formalin fixation.

Point 16:

Page 4 line 124

Response 16: The references have been added. [14]

Point 17:

Page 4 line 132

Response 17: The references have been added. [14, 15]

Point 18:

Page 4 line 134 reference

Response 18: The references have been added. [13,16, 17]

Point 19:

Page 4 line 141 the

Response 19: The word has been changed to a.

Point 20:

Page 4 line 142

Response 20: The antibody has been revised by antibodies.

Point 21:

Page 4 line 143

Response 21: Here has been added the “to be”.

Point 22:

Page 4 line 145,146

Response 22: the sentence has been revised. A study in Europe describing the prevalence and risk factors associated with T. gondii infection in wild birds found 19% ospreys surveyed to be seropositive.

Point 23:

Page 4 line 147

Response 23: Here has been added the infection.

Point 24:

Page 4 line 148 exposure rates

Response 24: The exposure rates have been edited.

Point 25:

Page 4 line 155

Response 25: The sentence has been added. They could also be infected by ingestion of oocysts present within the gastrointestinal tract of fish.

Point 26:

Page 4 line 158 one

Response 26: This has been changed to “a single”.

Point 27:

Page 4 line 160 may prey

Response 27: The sentence has been revised.

Point 28:

Page 4 line 162 This suggests that

Response 28: The sentence has been revised.

Point 29:

Page 4 line 170 concentrated in GI tract Uptake and transmission of Toxoplasma gondii oocysts by migratory, filter-feeding fish Massie et al.

Response 29: The reference is added.

Point 30:

Page 4 line 174 Italics

Response 30: T. gondii is changed.

Point 31:

Page 4 line 181 color?

Response 31: The swollen has been changed into enlarged. The color is dark red and has been added.

Point 32:

Page 5 line 184 Consider indicating tachyzoites with arrow or star to make it easier for readers to identify

Response 32: Thank you for your suggestions. The arrows have been added.

Reviewer 2 Report

1. Summary of content:
This is a well-written case report on a uncommonly reported disease in raptors.

2. Strengths and weaknesses:
It adds to the current literature on the disease, although there were not a lot of novel aspects compared to previously reported cases in birds and mammals. The thorough histopathological descriptions will be useful for practitioners encountering similar cases.

3. Point-by-point list of your recommendations for the improvement of the manuscript:
a. Figure 1 - a scale bar would be ideal to indicate the extent of any organomegaly.
b. Scale bars are needed. There is a black arrow in the insert of Figure 2a but there is no reference to it in the caption.

Author Response

Point 1: Summary of content: This is a well-written case report on an uncommonly reported disease in raptors.

Response 1: Thank you so much!

Point 2: Strengths and weaknesses: It adds to the current literature on the disease, although there were not a lot of novel aspects compared to previously reported cases in birds and mammals. The thorough histopathological descriptions will be useful for practitioners encountering similar cases.

Response 2: Response: I really appreciate your valuable suggestion. Indeed, this is the first case in raptor that will be useful for diagnosis similar cases.

Point 3: Point-by-point list of your recommendations for the improvement of the manuscript:
a. Figure 1 - a scale bar would be ideal to indicate the extent of any organomegaly.

Response: The scale bar is added in the pictures. The length of the white bar at the right bottom is 1-cm.

b. Scale bars are needed. There is a black arrow in the insert of Figure 2a but there is no reference to it in the caption.

Response 3: The scale bar and the arrowhead are added in the pictures (2a and 2b, insert). The information of the black bar at the right bottom has been added in the caption, thank you. 

Round 2

Reviewer 1 Report

Very nice.  Just a few grammatical corrections on the attached document 

Author Response

Point 1: Very nice.  Just a few grammatical corrections on the attached document

Response 1: Thank you for your time helping us with the editing and giving us suggestions on this manuscript.

Point 2: Page 2 line 58

Response 2: The “were” is deleted, thank you.

Point 3: Page 4 line 162

Response 3: This “suggest” is changed to “suggests”, thank you.

Point 4: consider arrows on the low magnification images showing intralesional tachyzoites in spleen.

 Response 4: Arrows are added in the H&E image, and the description is noted as "Photomicrograph of spleen showing severe multifocal to coalescing necrosis with intralesional Toxoplasma tachyzoites (arrow). H&E."  Thank you.